# Perturbation-Based Modeling Unveils the Autophagic Modulation of Chemosensitivity and Immunogenicity in Breast Cancer Cells

**DOI:** 10.3390/metabo11090637

**Published:** 2021-09-18

**Authors:** Isaac Quiros-Fernandez, Lucía Figueroa-Protti, Jorge L. Arias-Arias, Norman Brenes-Cordero, Francisco Siles, Javier Mora, Rodrigo Antonio Mora-Rodríguez

**Affiliations:** 1Research Center for Tropical Diseases (CIET), Laboratory of Tumor Chemosensitivity (LQT), Faculty of Microbiology, University of Costa Rica, San José 11501-2060, Costa Rica; isaac.quirosfernandez@ucr.ac.cr (I.Q.-F.); lucia.figueroa@ucr.ac.cr (L.F.-P.); jorgeluis.arias@ucr.ac.cr (J.L.A.-A.); norman.brenes@ucr.ac.cr (N.B.-C.); francisco.siles@ucr.ac.cr (F.S.); javierfrancisco.mora@ucr.ac.cr (J.M.); 2DC Laboratory, Laboratory of Surgery and Cancer, University of Costa Rica, San José 11501-2060, Costa Rica; 3Master’s Program in Microbiology, University of Costa Rica, San José 11501-2060, Costa Rica; 4Dulbecco Laboratory Studio, Residencial Lisboa 2G, Alajuela 20102, Costa Rica; 5Pattern Recognition and Intelligent Systems Laboratory (PRIS-Lab), Department of Electrical Engineering and Postgraduate Studies in Electrical Engineering, Universidad de Costa Rica, San José 11501-2060, Costa Rica

**Keywords:** autophagy, immunogenic cell death, chemotherapy, perturbation-based modeling

## Abstract

In the absence of new therapeutic strategies, chemotherapeutic drugs are the most widely used strategy against metastatic breast cancer, in spite of eliciting multiple adverse effects and having low responses with an average 5-year patient survival rate. Among the new therapeutic targets that are currently in clinical trials, here, we addressed the association between the regulation of the metabolic process of autophagy and the exposure of damage-associated molecular patterns associated (DAMPs) to immunogenic cell death (ICD), which has not been previously studied. After validating an mCHR-GFP tandem LC3 sensor capacity to report dynamic changes of the autophagic metabolic flux in response to external stimuli and demonstrating that both basal autophagy levels and response to diverse autophagy regulators fluctuate among different cell lines, we explored the interaction between autophagy modulators and chemotherapeutic agents in regards of cytotoxicity and ICD using three different breast cancer cell lines. Since these interactions are very complex and variable throughout different cell lines, we designed a perturbation-based model in which we propose specific modes of action of chemotherapeutic agents on the autophagic flux and the corresponding strategies of modulation to enhance the response to chemotherapy. Our results point towards a promising therapeutic potential of the metabolic regulation of autophagy to overcome chemotherapy resistance by eliciting ICD.

## 1. Introduction

Breast tumors are the leading cause of cancer-related death per year in the female population [1,2]. According to data from the National Cancer Institute of the United States of America, the 5-year survival rate of patients with localized breast tumors is close to 99%, but once it metastasizes to distant tissues, it decreases to 26% despite treatment [3]. Currently, the most commonly cytotoxic antineoplastic drugs used to treat breast cancer are paclitaxel, epirubicin and cisplatin [4]. It is well known that chemotherapeutic approaches have the disadvantage of damaging non-tumor cells throughout the body, hence eliciting multiple adverse effects [5]. In addition, the efficacy of these treatments is limited by the emergence of multi-drug resistance, such as increased drug transport, intracellular compartment drug sequestration, activation of detoxifying enzymes, suppression of apoptosis and the direct effect of the tumor microenvironment (physical barrier and detoxification) [6]. Nevertheless, in the absence of new therapeutic strategies, cytostatic drugs continue to be the most widely used treatment against metastatic breast cancer. This scenario highlights the imperative need to explore new therapeutic approaches for all types of breast cancer, which is definitely a hot research spot nowadays. Among the new therapeutic targets that are currently studied, we address here the regulation of autophagy and the induction of immunogenic cell death (ICD), which, in spite of being related, have not been thoroughly considered to design combined therapeutic approaches.

In the past decade, the dichotomy of apoptosis (tolerogenic)—necrosis (immunogenic) cell death has been left behind and, instead, cell death is classified in a broad spectrum of possible subroutines, implying that each has an associated degree of immunogenicity [7]. Immunogenicity of cell death relies on a combination of antigenicity, provided by neo-epitopes, and adjuvanticity, which depends on the release or exposure of damage-associated molecular patterns (DAMPs) [8]. Cancer cells, besides the reduction of antigenicity, avoid immune recognition by averting the release of DAMPs upon death. Hence, new therapeutic approaches aim to reinvigorate effective anti-tumor immune responses through the induction of ICD. In order to be considered as an ICD inducer, a treatment must activate cellular pathways associated with the release or exposure of DAMPs, such as endoplasmic reticulum stress response, autophagy and cell membrane permeabilization [8]. The endoplasmic reticulum stress response involves the unfolded protein response (UPR), which leads to down-regulation of the AKT protein and the mammalian target of rapamycin (mTOR) [9] and, consequently, to the translocation of calreticulin to the plasma membrane [10]. In addition, the activation of autophagy prior to cell death promotes the extracellular release of nucleotides such as ATP or UTP [11]. Calreticulin expression and extracellular ATP are recognized as DAMPs by immune cells and, therefore, are used as ICD markers. Another DAMP regularly used as an ICD marker is the release of the intranuclear protein high mobility group box 1 (HMGB1) [8]. Among chemotherapeutic drugs, anthracyclines (such as epirubicin) and oxaliplatin have been shown to induce ICD [12]. However, this depends on the type of tumor and its microenvironment. Conversely, in the absence of ICD markers, cell death is considered tolerogenic. Indeed, in breast cancer samples, a decreased autophagy or a reduced expression of HMGB1 were associated with a worse prognosis [13].

Autophagy is a highly regulated catabolic process that encompasses the lysosomal degradation of cytoplasmic components. There are three different types: (1) Microautophagy, in which lysosomes derive membrane invaginations to capture cytoplasmic material; (2) Chaperone-mediated autophagy, in which proteins with a specific pentapeptide signal are recognized by the heat shock protein 70 (HSP70) and subsequently translocated into the lysosome; and (3) Macroautophagy (hereinafter called “autophagy”), where cytoplasmic content, including organelles, is engulfed in double-membrane vesicles denominated autophagosomes (AP) that are fused with lysosomes in order to become autophagolysosomes (AL), the cellular microcompartment where degradation occurs [14]. Autophagy can be constitutive, when it involves the removal of damaged or senescent organelles in order to maintain basal energy balance, or adaptive when it occurs in response to starvation as a means to harness nutrients and sustain basic metabolic cell pathways by recycling intracellular components; thus, autophagy is essentially a metabolic process that regulates energy balance and, consequently, its dysregulation has been associated with many metabolic disorders [15].

In cancer, autophagy has a complex context-dependent role since it is associated both with tumor promotion and tumor suppression, varying among different models, types of cancer and its associated tumor microenvironment [16]. On one side, increased autophagy provides cancer cells with metabolic plasticity, and allows them to thrive in a nutrient-deprived environment [17], therefore, autophagy inhibition was proposed as an effective therapeutic strategy in advanced cancer. In addition, one of the main mechanisms cells use to adapt to stress is increasing autophagy and recent studies have suggested chemotherapy-induced autophagy as a response to stress that promotes drug resistance, but with autophagy inhibitors cancer cells can be sensitized to chemotherapy [18]. On the other hand, many tumor cells have mutations that make them resistant to apoptosis, hence, they can be exposed to considerable stress and damage without inducing cell death. Although controversial, many authors have previously proposed a regulated form of cell death called autophagic cell death, in which there is an excessive increase in autophagy that ends up killing the cells. Furthermore, cells resistant to apoptosis tend to die with morphological features that resemble an autophagic cell death, such as a high number of double-membrane vesicles. Therefore, the induction of autophagy as co-treatment with chemotherapy could facilitate autophagic cell death in tumor cells [19]. Currently, there are several clinical trials evaluating different pharmacological modulators of autophagy as a potential combinatorial treatment against cancer [20].

Moreover, there is evidence of the importance of autophagy in the exposure/secretion of ICD associated DAMPs. For instance, ATP secretion prior to cell death was associated with the autophagic flux [21,22,23], early-stage autophagy inhibitors down-regulate calreticulin translocation to the plasma membrane, while late-stage autophagy inhibitors rather promote its translocation [24], and HMGB1 seems to promote autophagy, thus, although autophagy does not facilitate its release, this protein could promote the secretion/exposure of other DAMPs through autophagy activation [25]. All this data suggests that ICD associated DAMPs have a significant correlation with the autophagic pathway and so we propose autophagy regulation as a new rationale to modulate the balance of immunogenic/tolerogenic cell death in favor of cancer treatment.

Precisely, we wondered if there is a specific autophagic phenotype as a determinant of the immunogenicity of cell death subroutines. The identification of such phenotype could orientate the design of combined therapeutic strategies of autophagic flux modulation and chemotherapeutic agents. Furthermore, it is important to ascertain if those modes of action hold true for different types of tumor cells with different basal autophagy levels and different response profiles to autophagic flux modulation agents. Therefore, we propose here a multiparametric study to describe the effect on cytotoxicity and immunogenicity through the modulation of the autophagic flux in cells treated with chemotherapy.

In the study of autophagic flux, in vitro approaches have focused on LC3 protein fluorescent sensors, given that LC3 is specifically incorporated into the autophagosomal membranes. Upon pro-autophagic stimuli, such as nutrient deprivation, the LC3 sensor redistributes from a diffuse (LC3-I) to a clustered pattern (LC3-II), due to the formation and agglomeration of AP. In this sense, it is important that not all LC3-II is present on autophagic membranes and some population of LC3-II seems to be ectopically generated in an autophagy-independent manner, therefore measuring the autophagic flux is a better approach to precisely evaluate the autophagic activity [26,27].

Autophagic flux can be monitored by the quantification of fluorescent puncta on a single cell basis as we previously did [28]. However, this approach has the limitation that the clustering pattern of the APs can change significantly upon perturbations leading to autophagic vesicles of varying sizes and intensities. For instance, when AP-lysosome fusion is blocked, larger APs are detected, possibly due to AP-AP fusion, or to an inability to resolve individual APs when they are present in large numbers. Although it is possible to detect changes in the size of puncta by fluorescence microscopy, it is not possible to correlate size with autophagy activity without additional assay methods. Size determinations are also problematic and size estimation on its own without considering puncta number per cell is not recommended as a method for monitoring autophagy. However, it is possible to quantify the fluorescence intensity of GFP-Atg8- family proteins at specific puncta, which does provide a valid measure of protein recruitment [29].

In addition, the single LC3-GFP approach does not enable the differentiation between AP and AL, therefore, novel sensors with two fluorescent sensors associated with LC3 in tandem were devised. As such, one sensor fluorescence is quenched at low pH (e.g., GFP) while the other is stable (mCherry) at this condition. Using these validated sensors of autophagic flux, it is possible to identify whether a phagosome has already been fused or not with a lysosome [30]. Still, the autophagic pathway is very complex and the understanding of the combinatorial effects of chemotherapeutic drugs on the autophagic metabolic pathway is not straightforward. Therefore, robust image analysis pipelines, mathematical modeling and systems biology approaches are the best choice to decode these complex biological interactions.

Systems biology acts as a hatchet to unveil the underlying principles governing biological processes through the definition of how its components are organized and interlinked. By implementing various algorithms to analyze a network, the core modulators can be detected, and the dynamics of these core sets of modulators can be studied through mathematical modeling [31]. These strategies were used to assess autophagy in the context of neurodegenerative diseases [32], endoplasmic reticulum stress [33], lung cancer [34], and many other physiological and pathological processes [31]. Moreover, in perturbation-based models, changes in the outputs are attributed to perturbed inputs and used to estimate their importance for a particular instance. In addition, these methods can explain the model’s decision for each individual predicted instance as well as for the model as a whole [35]. In addition, perturbation-based methodologies have shown to be useful in the prediction of cancer chemosensitivity [36].

In the present study, we report the implementation and validation of an image analysis platform to describe the autophagic flux in breast cancer cells using the mCHR-GFP tandem LC3 sensor. The ratio of AL/AP reports dynamic changes of autophagic flux in response to starvation-induction and bafilomycin-mediated degradation blockade. These breast cancer cell lines display heterogeneous basal levels of autophagy and a differential response to drugs inducing known perturbations of autophagic flux. The perturbation-based dynamic modeling of the autophagic sensor indicates cell-line specific modes of action of chemotherapeutic agents on the autophagic flux. This study represents a first approach to explore the influence of autophagy in tumor chemotherapy-induced cytotoxicity and immunogenicity through the application of systems biology and perturbation-based models. Our results point out a promising therapeutic potential of the regulation of autophagy to overcome chemotherapy resistance by eliciting ICD.

## 2. Results

### 2.1. The Autophagolysosome/Autophagosome Ratio of an mCHR-GFP Tandem LC3 Sensor Reports Dynamic Changes of Autophagic Flux in Response to Starvation-Induction and Bafilomycin-Mediated Degradation Blockade in Breast Cancer Cell Lines

The use of tandem fluorescent sensors is the most widely used technique for monitoring autophagy since it allows for the quantification of two different cellular compartments simultaneously. This approach takes advantage of the low resistance of GFP fluorescence to acidic pH environments in contrast to the stable fluorescence of mCherry to those environments [29]. To simulate that perturbation, a treatment condition with bafilomycin is normally included. Bafilomycin is a macrolide that recognizes and interacts with the V-ATPase, inhibiting the acidification of AL, however, by a poorly understood mechanism, it also causes an inhibition of the fusion of AP with lysosomes [37].

In order to monitor the autophagic flux in living breast cancer cells, we aimed to produce cell lines with stable expression of the mCHR-GFP-LC3 tandem sensor. To achieve this, the gene construct was introduced into three different breast cancer lines using the triple transduction procedure. Briefly, the vector was assembled into HEK293T cells by the transduction of the pBABE-pure plasmids mCherry-EGFP-LC3B, pMD2.G and pCL-Eco with polyethyleneimine [38]. The vesicular stomatitis virus (VSV) glycoprotein was introduced using the pMD2.G to improve retroviral tropism and transduction efficiency. Breast cancer cell lines MCF-7, T-47D and MDA-MB-468 monolayers were transduced with retroviral particles carrying the mCherry-EGFP-LC3B construct. Transduced cells were selected using puromycin and separated using fluorescence-activated cell sorting. Subsequently, we determined the kinetic changes in this molecular sensor upon treatment with drugs inducing known perturbations of the autophagic flux by a kinetic live-cell imaging assay. For this, we designed a pipeline to separate and quantify APs (mCherry-positive/GFP-positive) and ALs (mCherry-positive/GFP-negative) (Appendix A).

Cells in the supplemented culture medium did not show any evident changes in the distribution and intensity of the sensor. However, when treated with bafilomycin, an accumulation of white fluorescent puncta was observed over time. This accumulation represents the basal autophagic flux of each cell line upon degradation blockade of AP (Figure 1A). On the other hand, cells in starvation medium showed a dramatic loss of green fluorescence, suggesting an increase in the synthesis of new AP in parallel to an increase in the fusion with lysosomes, leading to AL formation. After bafilomycin blockade, the accumulation of AP was greater than in basal conditions but differences were observed among those cell lines (Figure 1A,B). Indeed, we chose these three cell lines to cover a broad range of autophagic phenotypes including a cell line with high basal autophagic flux (T-47D in complete medium), a cell line with low basal autophagic flux but a high response to starvation (MCF-7) and a cell line with no basal autophagic flux and a lower response to starvation (MDA-MB-468).

### 2.2. Breast Cancer Cell Lines Display a Heterogeneous Response to Drugs That Induce Known Perturbations of Autophagic Flux

In order to confirm that the mCHR-GFP-LC3 sensor expressed in our breast cancer cell lines is able to report autophagic flux perturbations, we assessed the effect of several compounds which were extensively described as autophagy modulators. This includes the mTORC 1 inhibitor rapamycin, the mTORC1/2 inhibitor Torin 2, the USP10/13 inhibitor Spautin, the PI3K inhibitor Wortmannin and the lysosomotropic compound hydroxychloroquine (HCQ). All of these compounds are widely used as autophagic flux regulators in vitro and in vivo and were studied as stand-alone or combined therapies against different types of tumors. For this reason, we evaluated the changes over time reported by the autophagy sensor upon treatment of breast cancer cell lines with different concentrations of these drugs through a kinetic fluorescence microscopy assay. Autophagy inhibitors were evaluated in complete and starvation medium to assess the effect of these substances over basal and induced autophagy respectively, whereas the autophagic inducers were evaluated in complete medium only. The concentration of each autophagic modulator was selected taking into account the toxicity 72 h after treatment (data not shown).

Rapamycin treatment caused a clear change in the cellular distribution of the sensor fluorescence for all three cell lines (Figure 2A, RAP). When compared with untreated cells, APs become very scarce, opposed to ALs, which increase in intensity and number of puncta. These visual variations translate into an increment in AL/AP ratio after image analysis. Torin-2 caused a much more intense effect than rapamycin (Figure 2A, TOR) resembling the sensor relocation observed under starvation conditions (Figure 2B, untreated control), which hints that both perturbations have a similar effect on this pathway. The sensitivity to the induction of autophagy by treatment with mTOR inhibitors was higher in MCF-7 and T-47D lines when compared to the MDA-MB-468 cell line, similar to what was observed in their basal and starvation-induced autophagy.

Spautin-1 produced changes in sensor intensity in all three cell lines. There was a decrease in both APs and ALs, which was appreciable in complete medium (Figure 2A, SPA) and under starvation conditions (Figure 2B, SPA). The decrease in APs was also higher leading to an increase in AL/AP ratio. Similarly, Wortmannin caused a decrease in the intensity of both APs and ALs. However, when applied in starvation medium, an early-stage autophagy blockade occurred even at the lowest concentration tested. This behavior was reflected in the inability of the sensor to relocate to clusters (Figure 2B, WRT). Finally, hydroxychloroquine caused the accumulation of APs, an effect that becomes even more noticeable in nutrient-deprived medium (Figure 2B, HCQ). This perturbation is perceived as a decrease in AL/AP ratio.

The results obtained with all these autophagy modulators are consistent with the expected regulator activity previously reported for each of them. This further demonstrates the ability of our sensor to differentially report a wide variety of known perturbations to autophagy, which proves to be heterogeneous among different breast cancer cell lines.

### 2.3. Cell Line-Specific Profiles of Cytotoxicity and Cell Death Immunogenicity Arise from the Interactions between Autophagy Regulators and Chemotherapeutic Perturbations

The gold standard strategy to categorize cellular death as immunogenic relies on in vivo tests involving the vaccination of immune-competent mice with syngeneic cancer cells after they were treated in vitro with the drugs to be evaluated. Nonetheless, this approach cannot be used to test human cancer cell lines and is expensive and difficult to implement as a screening test. An alternative method to assess immunogenic cell death consists of the determination of DAMPs secreted or exposed by cells prior and during cell death. Traditionally, calreticulin exposure to the plasma membrane, extracellular ATP secretion and extracellular HMGB1 release were determined for this purpose [8].

In order to evaluate the secretion/exposure of immunogenic cell death associated DAMPs exposed or released by the interaction of autophagy regulators and chemotherapeutic drugs, we performed a cytotoxicity assay in which breast cancer cells were treated with chemotherapeutics drugs alone, autophagy regulators alone and all the possible combinations of those groups of drugs. After 24 h of treatment, calreticulin exposure was determined on live cells, and after 72 h of treatment, the live-cell percentage was measured along with the ATP concentration in the culture supernatant. We compared these immunogenicity profiles with the changes observed in AL, AP and the AL/AP ratio at 48 h post-treatment. In general, the autophagy regulators had minimal effects on cytotoxicity (Figure 3A and Appendix A), calreticulin exposure (Figure 3B) and ATP secretion (Figure 3B) when used alone; however, these regulators modified those outputs when used in combination with chemotherapeutic drugs.

Regarding paclitaxel, in the MCF-7 cell line, this drug displayed a strong synergism with rapamycin, and a slighter synergism with Torin-2, HCQ and wortmannin. From those interactions, rapamycin/HCQ also caused an increment in calreticulin exposure. In the T-47D cell line, there were no evident synergistic interactions, and HCQ and Spautin-1 practically nullified the cytotoxic activity of this drug. Nonetheless, Torin-2 and wortmannin did potentiate calreticulin exposure while HCQ caused an increase in ATP secretion despite antagonizing cell death and calreticulin exposure. In the MDA-MB-468 cell line, paclitaxel did not show any strong interactions for cytotoxicity with autophagy modulators. However, rapamycin and HCQ led to an increase in calreticulin exposure although no combination elicited a greater ATP secretion than paclitaxel alone. It is of note that paclitaxel-induced ATP secretion in all three cell lines occurred, with strong increments when used in combination with HCQ in MCF-7 and T-47D cell lines. On the other hand, Spautin-1 antagonized the cytotoxic effect of paclitaxel, which was also reflected by a decrease in calreticulin and ATP levels for that condition in all three cell lines.

In respect of cisplatin, in the MCF-7 and T-47D cell lines, the impact of autophagy modulators on cisplatin cytotoxicity was similar, with a less pronounced antagonism when treated together with rapamycin and Spautin-1 in the T-47D cell line. Furthermore, in both these cell lines, no ATP secretion was detected in any cisplatin combination and there were no concomitant appreciable interactions regarding calreticulin exposure, except for HCQ which potentiated calreticulin exposure in the MCF-7 cell line, in spite of having no impact on cisplatin cytotoxicity. Conversely, in the MDA-MB-468 cell line, cisplatin displayed remarkable synergisms with rapamycin, Torin-2 and HCQ, although, Spautin-1 antagonized cisplatin-induced cell death, as observed with the other two cell lines. Additionally, HCQ potentiated cisplatin-induced calreticulin exposure and was the only autophagy modulator that enabled ATP secretion when combined with cisplatin.

With respect to epirubicin, in the MCF-7 cell line, rapamycin showed a slight synergism and Spautin-1 displayed a strong antagonism. As for calreticulin determination, no combination elicited a higher exposure than epirubicin alone and Spautin-1 actually completely blocked calreticulin translocation, an expectable outcome due to the strong antagonism in cytotoxicity. In the T-47D cells, results were different as shown by an antagonism in cytotoxicity with rapamycin but an evident synergism with HCQ and, once again, a strong antagonism with Spautin-1. Despite cytotoxicity antagonism, rapamycin and Spautin-1 caused an increase in calreticulin exposure compared to the epirubicin treatment alone. Cytotoxicity interactions with the MDA-MB-468 cell line were similar to those found in the MCF-7 cell line, however, rapamycin and wortmannin induced a higher exposure of calreticulin, compared to epirubicin alone. Lastly, ATP secretion was increased in epirubicin and HCQ combined treatment for all three cell lines, highlighting the importance of this combined treatment, despite cytotoxicity interactions and calreticulin translocation.

These results confirm that cytotoxicity and immunogenic cell death associated with DAMP exposure/secretion interaction profiles with autophagy regulators are cell line-specific and that the same combination of drugs might be synergistic in one cell line and antagonistic in another cell line.

### 2.4. Perturbation-Based Dynamic Modeling of the Autophagic Sensor Indicates Cell-Line Specific Modes of Action of Chemotherapeutic Agents on the Autophagic Flux

In addition to being a physiological metabolic process, autophagy serves as a response to different types of cellular stress such as hypoxia, nutrient deprivation, or stress-induced by chemotherapeutic compounds. Moreover, autophagy plays an active role in the determination of multiple cell death subroutines [7]. Since chemotherapy is pleiotropic, the use of classical approaches to identify the precise effect of each drug over the autophagic flux for each tumor implies a challenge when designing personalized cancer therapies. In order to explore the ability of the mCHR-GFP-LC3 sensor to report autophagic flux qualitative changes induced by chemotherapy, we proceeded to expose cells to three different chemotherapeutic drugs at a concentration that caused approximately 50% of cytotoxicity after 72 h. To evaluate the effect on autophagy in stages prior to cell death, we monitored the mCHR-GFP-LC3 sensor between 36 and 48 h upon addition of chemotherapy to cell cultures.

The results showed evident changes in the sensor fluorescence distribution and intensity when compared to the untreated control (Figure 4A). Epirubicin treatment caused a slight decrease in GFP fluorescence intensity in all three cell lines, nonetheless, there were differences between cell lines regarding the mCherry fluorescence. In the MCF-7 and MDA-MB-468 cell lines there were no clear changes, while in the T-47D cell line, this chemotherapy induced the formation of large ALs clusters. On the other side, upon cisplatin treatment, there was a dramatic increase in the amount and size of ALs in the MCF-7 and the T-47D cell lines, whereas this same chemotherapy induced an increase in the amount of small-sized ALs with a concomitant slight decrease in the quantity of APs in MDA-MB-468 cell line. Finally, when exposed to paclitaxel, GFP fluorescence remained diffuse in the cytoplasm in the MCF-7 and T-47D cell lines. Additionally, in the MCF-7 cells, small ALs were observable in large numbers throughout cells. Conversely, this chemotherapy produced a marked decrease in APs and ALs in the MDA-MB-468 cell line.

These results suggest that the sensor is able to report changes in autophagic flux induced by chemotherapy. Therefore, to infer the specific mode of action of each chemotherapeutic perturbation on the autophagic flux, we performed double treatments of chemotherapy and drugs with known mechanisms of action on the autophagic machinery.

Quantitative measurement of APs and ALs for all the double perturbations mentioned above confirmed that the variations in autophagy induced by these chemotherapeutic compounds are cell line-specific. The perturbation-response profiles for a specific treatment condition differed among the three cell lines (Appendix A), hindering the direct interpretation of a possible mode of action for each chemotherapeutic treatment.

Given the large amount of data obtained and its high complexity, a systems biology approach was required. Hence, we developed a family of mathematical models representing alternative hypotheses of the mode of action of each chemotherapeutic drug over the different steps of the autophagic flux. This data-driven approach has the potential to help us identify the model with the modes of action that better fit the experimental data for each breast cancer cell line. 

We started by proposing a network topology of the mCHR-GFP-LC3 sensor metabolic pathway. Using this basal topology, we introduced variations considering different single modes of action for each chemotherapeutic drug and all combinations thereof. Afterward, the family of network topologies was converted into a family of SBML mathematical models using CellDesigner. The SBML models were imported in COPASI to perform the parameter estimation task for each of them. The lower objective function value of the parameter estimation was the metric to indicate the fitness of each model to the experimental data.

The best-fitted model for each cell line provides insight regarding which combination of modes of action is the most congruent with the experimental perturbation data (Figure 4B). Our model suggests that epirubicin treatment increases the degradation of ALs in the MCF-7 cell line, whereas this anthracycline stimulates the formation of ALs in the T-47D cell line. On the other hand, it appears that this drug inhibits the formation of new APs in the MDA-MB-468 cell line. Regarding cisplatin, our model proposes that this drug inhibits the degradation of ALs in the MCF-7 and T-47D cell lines, while it seems to block the production of new ALs in the MDA-MB-468 cell line (Figure 4B). Finally, in our model, paclitaxel blocks the degradation of APs in the MCF-7 and T-47D cell lines, whereas it stimulates the degradation of ALs in the MDA-MB-468 cell line.

This data-driven approach indicates that the mCHR-GFP-LC3 sensor reports the modes of action of chemotherapeutic perturbations on the autophagic flux and suggests specific modes of action for epirubicin, cisplatin and paclitaxel across three different breast cancer cell lines.

## 3. Discussion

The formation of the AP comprises three basic steps: initiation, nucleation and elongation. The central metabolic regulator of autophagy is mTOR, which acts through two different complexes, mTOR complex 1 (mTORC1) and mTOR complex 2 (mTORC2). Both complexes require the presence and interaction of the proteins Deptor, Tel2, Tti1 and mLST8. Moreover, mTORC1 additionally requires the RAPTOR and PRAS40 proteins, while mTORC2 requires the Rictor and mSin1 proteins [39]. In the absence of autophagic stimuli, mTOR phosphorylates and inhibits the next effector of the pathway, the ULK complex, composed of ULK-1, ATG13, ATG101 and FIP200. During nutrient deprivation, the negative regulation that mTOR has on the ULK complex is suppressed, allowing it to be translocated to the endoplasmic reticulum membrane to initiate the phagophore formation [40]. Numerous compounds have been identified to be early-stage autophagic flux modulators. One of the first inducers used as a cancer treatment was rapamycin, a drug that inhibits mTORC1. To improve its pharmacological properties, rapamycin-like molecules called “rapalogs” were designed, which are currently used as a treatment for breast and kidney cancer. Other compounds, such as Torin-2, are potent inhibitors of both mTORC1 and mTORC2 [41]. Bortezomib is a proteasome inhibitor that induces autophagy as a compensatory mechanism and it is used as a treatment for multiple myeloma and mantle cell lymphomas [42].

After the initiation of autophagy, the next step is nucleation. In the endoplasmic reticulum membrane, the ULK complex activates another important complex, the phosphatidylinositol kinase type 3 complex (PI3K class 3), which is composed first of all by the vesicular protein sorting 34 kinase (Vps34), Vps15, the beclin-1 protein and finally the ATG14 protein. Once activated, this complex catalyzes the conversion of phosphatidylinositol to phosphatidylinositol triphosphate which is essential for the next step and to recruit the protein machinery necessary to continue the pathway [43]. Two widely used autophagy regulators, 3-methyladenine and wortmaninn, block this stage by inhibiting PI3K class 3 [44]. The specific autophagy inhibitor 1 (Spautin-1) also blocks nucleation by inhibiting ubiquitin-specific peptidases 10 and 13, which leads to beclin-1 degradation [45].

In the third phase of the autophagic flux, two ubiquitin-like conjugation systems function at a late step of AP function: ATG12-ATG5 and ATG8-phosphatidylethanolamine (PE). ATG12 is activated by ATG7 and then conjugated to ATG5. The enzyme ATG10 catalyzes this conjugation. The new protein ATG12-ATG5 is non-covalently associated with ATG16. On the other hand, ATG8 and its mammalian homologs the microtubule-associated protein 1 light chain 3 (MAP1LC3), also known as LC3, and the γ-aminobutyric acid type A (GABAA) receptor-associated protein (GABARAP) are synthesized as precursors with an additional sequence at C terminal end that is eliminated by proteases ATG4A-D, then LC3/GABARAP is conjugated to the amino group of the phosphatidylethanolamines present in the membrane of the phagophore that is expanding, and this conjugation is in charge of ATG7 and ATG3 activation, which leads to LC3/GABARAP up to the phospholipid. The aforementioned ATG12-ATG5-ATG16 complex participates actively in the conjugation of LC3/GABARAP to the phosphatidylethanolamine [40,46,47].

Once the AP is formed, it migrates through the cytoskeleton to the lysosome for the fusion to take place. It was speculated that the vacuolar ATPase that keeps the pH low within the lysosome served additionally as a receptor for the AP. This is because the use of an inhibitor of this ATPase such as bafilomycin-A2 causes AP-lysosome fusion blockade. However, the selective fusion mechanism of these two organelles is not known with certainty [37]. In this step, inhibitors of the autophagic flux were used as an important therapeutic target to overcome prosurvival mechanisms elicited by chemotherapy treatment. One of the approaches is the use of lysosomotropic molecules that inhibit lysosome acidification and AP-lysosome fusion, an example of these compounds are chloroquine and hydroxychloroquine, classical anti-malarial drugs. Other molecules with similar mechanisms of action are bafilomycin A2 [42] and artesunate [48]. Many of these drugs are currently in phase one or two of clinical trials as treatments against cancer [20].

Despite these advances in the modulation of autophagy, there is still a gap in the understanding of the interactions between chemotherapy and modulators of autophagy. Indeed, autophagy perturbing agents can inhibit or activate different steps of the pathway but their potential side effects must be considered in the interpretation of the secondary consequences of autophagy perturbation, especially in long-term studies as ours. For example, lysosomotropic compounds can increase the rate of AP formation by inhibiting MTORC1, as activation of lysosomally localized MTORC1 depends on an active V-ATPase. HCQ treatment may cause an apparent increase in the formation of APs possibly by blocking fusion with the lysosome as observed by the reduction of colocalization of LC3 and LysoTracker™ despite the presence of APs and lysosomes. However, it can also block AP-lysosome fusion in HeLa and MEFs but this mechanism could be cell-type specific, as other studies report that CQ prevents AL clearance and degradation of cargo content, but not AP-lysosome fusion [29].

Cellular metabolism and other physiological processes such as autophagy are highly complex, and classic mechanistic strategies to study biological phenomena are becoming less effective with the advent of technologies that engender a large amount of information. One way to infer the mechanism of action of an unknown perturbation on a metabolic pathway is to use perturbation-based modeling, a strategy in which simple modifications are introduced to a highly complex model followed by time-course monitoring of the effect upon the system. After accumulating data about the impact of many simple perturbations, as well as the combination of some of these, it is possible to mathematically model the effect of known perturbations, with the advantage that the model is robust enough to infer the effect of unknown perturbations [49].

After validating the mCHR-GFP tandem LC3 sensor capacity to report dynamic changes of the autophagic flux in response to external stimuli (Figure 1) and demonstrating that both basal autophagy levels and response to diverse autophagy regulators fluctuate among different cell lines (Figure 2), we explored the interaction between autophagy modulators and chemotherapeutic agents in regards of cytotoxicity and ICD using three different breast cancer cell lines (Figure 3 and Figure 4) and proposed specific modes of action of chemotherapeutic agents on autophagic flux modulation through perturbation-based dynamic modeling (Figure 4B). The three chemotherapeutic agents evaluated were the most commonly used treatments against breast cancer, paclitaxel, cisplatin and epirubicin.

Paclitaxel was able to induce ICD-associated DAMPs as a solo treatment in all three breast cancer cell lines, although this was more evident in the T-47D cell line. This induction was also reported through in vitro and in vivo models in ovarian cancer [50]. Interestingly, this exposure/release of ICD-associated DAMPs was exacerbated when paclitaxel was used in combination with some autophagy modulators, particularly with HCQ in the MCF-7 cell line. These results suggest that paclitaxel combined with HCQ could enhance ICD in some breast cancers, although this has to be studied in the future. Furthermore, many authors have previously demonstrated that autophagy inhibition in tumor cells reverses their resistance to paclitaxel [51,52,53]. This was related to the autophagy cytoprotective effect against stress [54], but our results with autophagy inhibitor HCQ suggest that this drug sensitization could also be related to the induction of ICD-associated DAMPs. Additionally, when treated with paclitaxel, breast cancer cells revealed changes in their autophagic flux and our perturbation-based model suggests that this drug blocks AL formation in some scenarios and increases AL degradation in others. This variation agrees with what has been previously reported since paclitaxel was described to have a pleiotropic effect on autophagy, which may be dependent on the expression level of autophagy initiation proteins. Some authors report that this drug may cause an inhibition of the activity of the PI3K class 3 complex or that it could inhibit the mobilization of AP towards lysosomes [55], while others describe this drug as a potent autophagy activator [56]. These contrasting results reveal that the paclitaxel effect on autophagy is context-dependent.

Cisplatin was not able to release/expose ICD-associated DAMPs in two of three breast cancer cell lines when exposed on its own, which was expected since this drug was previously categorized as a non-ICD inducer [57]. Surprisingly, cisplatin did induce ATP secretion and calreticulin exposure in the MDA-MB-468 cell line, which suggests that this drug could be an ICD inducer in some cases of low basal autophagy, but as we mentioned before, this has not been previously reported, so it needs further research. Moreover, breast cancer cells increased AP and AL formation when exposed to cisplatin and our model positions this drug mainly as a blocker of AL degradation. This agrees with previous reports showing activation of autophagy in tumor cells upon cisplatin treatment [58], which has been related as a protector factor from cell death [59] and further validated in some studies where concomitant use of autophagy inhibitors promotes cisplatin-induced cell death [60,61]. Contrariwise, in one of the breast cancer cell lines analyzed here, there was a strong cytotoxic synergism between cisplatin and autophagy inducers rapamycin and Torin-2, which again reinforces the idea that each cell line, and hence each type of breast cancer, is going to respond differently to regulators of the autophagic flux. It seems that in this cell line autophagy induction in the context of cisplatin induces autophagy, but not enough for this metabolic process to act as a protector factor form cell death.

As for epirubicin, it is recognized as a bona fide ICD inducer [12] and, remarkably, our results suggest that this property could be enhanced through autophagy modulators, specifically the double therapy of epirubicin+HCQ was one of the top inducers of ICD-associated DAMPs described here. With regards to the autophagic flux, epirubicin had variable effects in our model, which is consistent with previous reports. On one side, there is evidence that autophagy protects breast cancer cells from apoptosis induced by epirubicin facilitating resistance to this drug [62] and that autophagy inhibition increases epirubicin antitumor activity in breast cancer models [63]. On the other side, there are reports of tumor sensitization to epirubicin following the induction of autophagy with rapamycin-derived drugs [64].

Our perturbation-based approach indeed enabled us to propose alternative modes of action of the chemotherapeutic drugs on the autophagic flux. As discussed above, chemotherapy was reported to be pleiotropic and have a context-dependent effect on autophagy. Upon this conundrum of different activities for an individual chemotherapeutic drug, it becomes critical how the cell senses the stress signals elicited by chemotherapy. Therefore, the specific configurations of critical pathways such as the autophagy pathway are paramount in cell fate decisions. In order to validate these model-driven hypotheses for those specific configurations of autophagy and their association to chemosensitivity a larger functional genomics study could be conducted with cell lines to identify the critical proteins involved in each of those configurations and to identify the gene expression signatures that could be used to correlate these findings with the chemosensitivity of human tumors.

This approach also enabled us to look for a common autophagic phenotype related to the exposure or release of ICD-associated DAMPs. Some of the most potentially immunogenic combinations such as PAC+HCQ, PAC+RAP in the MCF-7 cells and PAC+TOR, EPI+RAP or EPI+SPA in T-47D cells have in common the accumulation of APs. This suggests that an active AP formation with an impaired fusion to the lysosome represents a potential autophagic phenotype related to increased cytotoxicity and immunogenicity. Future experiments using single-cell technologies should be designed to correlate DAMP exposure and sensors of autophagic flux to validate this hypothesis. In vivo experiments monitoring endogenous LC3 expression and clustering, and the characterization of the immune infiltrate by immunohistochemistry and flow cytometry may contribute to assess whether the accumulation of APs upon treatment correlates with ICD in order to postulate combinatorial strategies designed to favor this autophagic phenotype.

This study is the first to exploit systems biology and perturbation-based models in order to scan the association between autophagy and chemotherapy-induced cytotoxicity and immunogenicity. Our results point towards a promising therapeutic potential of the metabolic regulation of autophagy to overcome chemotherapy resistance by eliciting ICD. Furthermore, treatment combinations elicited diverse ICD markers induction and autophagic flux perturbations among the different cell lines, highlighting the importance of evaluating these therapeutic strategies according to the type of cancer.

## 4. Materials and Methods

### 4.1. Cell Lines

HEK293T cells (ATCC CRL-3216) were cultured in Dulbecco’s modified Eagle medium (DMEM, Gibco, New York, NY, USA, 10569044) supplemented with 10% heat-inactivated fetal bovine serum (FBS, Gibco 10438026) and 1X antibiotic-antimycotic solution (Gibco 15240062).

MCF-7, T-47D, and MDA-MB-468 breast cancer cell lines were obtained from the National Cancer Institute (NCI) collection and cultured in Roswell Park Memorial Institute medium (RPMI 1640, Gibco 11835030) supplemented with 10% FBS (Gibco), 1X GlutaMAX (Gibco 35050061), and 1X antibiotic-antimycotic solution (Gibco). All cell lines were cultured at 37 °C in 5% CO_2_.

### 4.2. mCherry-EGFP-LC3B Retroviral Vectors Assembly

Retroviral particles were generated in 60% confluent HEK293T cell monolayers by triple transfection with polyetherimide (Polysciences 23966) of the transfer plasmid pBABE-puro mCherry-EGFP-LC3B [65] (a gift from Jayanta Debnath, Addgene plasmid #22418), the packaging plasmid pCL-Eco (a gift from Inder Verma, Addgene plasmid #12371) and the envelope plasmid pMD2.G (a gift from Didier Trono, Addgene plasmid #12259), as previously described [66]. At 72 h post-transfection, virus-containing media was collected, filtered through a 0.45 µm membrane, supplemented with 5 µg/mL polybrene (Sigma, Kawasaki, Japan, H9268), and stored at −80 °C.

### 4.3. Generation and Sorting of Stable mCherry-EGFP-LC3B Cell Lines

MCF-7, T-47D and MDA-MB-468 cells monolayers at 80% confluency were transduced with retroviral particles carrying the mCherry-EGFP-LC3B construct and centrifuged for 2 h at 1500 rpm, 25 °C. At 72 h post-transduction, cells were selected with 8 μg/mL puromycin (Sigma P8833) in RPMI 1640 10% FBS for 2 days. Cell sub-populations with homogeneous levels of expression of the mCherry-EGFP-LC3B construct were isolated for each cell line by fluorescence-activated cell sorting in a BD FACSJazz™ cell sorter (BD Biosciences, Franklin Lakes, NJ, USA). Sorted cells were cultured in RPMI 1640 10% FBS with 2 µg/mL puromycin.

### 4.4. Autophagy Modulators and Chemotherapeutic Drugs

Autophagy activators rapamycin (Sigma-Aldrich, St. Louis, MO, USA, R0395) and Torin-2 (Sigma-Aldrich SML1224) were dissolved in dimethyl sulfoxide (DMOS, Sigma-Aldrich D8418). Autophagy inhibitors wortmannin (Sigma-Aldrich W1628), bafilomycin A1 (Sigma-Aldrich B1793) and Spautin-1 (Sigma-Aldrich SML0440) were dissolved in DMSO (Sigma Aldrich, St. Louis, MO, USA, K3753), while the hydroxychloroquine sulfate salt (Sigma Aldrich H0915) was dissolved in deionized water.

Epirubicin and cisplatin were a gift from the Oncologic Pharmacy of the Calderon Guardia Hospital from the Social Security System of Costa Rica, as aqueous and DMSO solutions, respectively. Paclitaxel was dissolved in DMSO (Sigma-Aldrich T7402).

MCF-7 cell line was treated using the following chemotherapeutic drugs concentrations: epirubicin 1.70 µM, cisplatin 12.5 µM and paclitaxel 7.70 µM. The T-47D cell line was treated using the following drug concentrations: epirubicin 0.85 µM, cisplatin 25.0 µM and paclitaxel 7.7 µM. MDA-MB-468 cell line was treated using the following drug concentrations: epirubicin 0.85 µM, cisplatin 1.60 µM and paclitaxel 7.7 µM. These concentrations are similar to those reached by these drugs in the intratumoral microenvironment and used by cytotoxicity assays [67]. Drug concentrations were individualized for each cell line to attain cell mortality of 40–50% for each treatment.

### 4.5. Autophagic Flux Kinetic Assay

Breast cancer cell lines with stable expression of the mCherry-EGFP-LC3B sensor were seeded in 96-well plates (Greiner Bio-One, Kremsmünster, Austria, µclear 655090), after 24 h incubation the medium was replaced with one containing the respective treatments. Images were taken in the green (GFP) and red (mCherry) channels every 3 h for 12 h in the Cytation 3™ (Biotek, Winooski, VT, USA) automated fluorescence microscope. In order to induce autophagy, the Krebs-Henseleit (Sigma Aldrich) starvation medium was used, and bafilomycin A1 (Sigma Aldrich) was used at 50 nM to cause a blockade in the autophagic flux.

### 4.6. Live Cell Image Analysis

In order to quantify AP and AL in a time-resolved manner, an image analysis protocol was made in Cell Profiler software (ver 3.0.0). It was adjusted for the automatic segmentation of cells and the quantification of mCherry positive GFP positive AP and mCherry positive GFP negative AL.

### 4.7. Cytotoxicity Assay

Cell death was determined using a Hoechst 33,342 (Invitrogen™, Waltham, MA, USA, H3570) and propidium iodide (PI, Invitrogen™ P3566) stain assay. Cells were seeded in 96 well plates (Greiner Bio-One µclear), incubated for 24 h and then treated for 72 h with a pre-established concentration of each chemotherapeutic treatment alone, autophagy modulator alone, or combinations of those. Once the incubation time had elapsed, Hoechst (1.25 µg/mL final concentration) and propidium iodide (10 µg/mL, final concentration) were added to the medium of each well. Images in the red (PI, dead) and blue (Hoechst, all nuclei of all cells) channels were taken. With this information, the live-cell percentage was determined using the Cell Profiler image analysis software, through an image analysis pipeline designed to count total and dead cells. Live cells were calculated by dividing PI negative cells by the total amount of cells in each well. Cell death interactions (synergism vs. antagonism) were estimated by comparing the multiplied effect of each treatment alone with the real effect elicited by the combined treatment.

### 4.8. Calreticulin Measurement

Plasma membrane Calreticulin exposure was determined using a monoclonal anti-human calreticulin allophycocyanin (APC)-conjugated antibody (R&D Systems, Minneapolis, MN, USA, IC3898a). Cells were seeded in 24 well plates and incubated for 24 h, afterwards the medium was replaced with a new medium containing the different treatments. After 24 h of incubation, the medium was removed and each well was washed using phosphate-buffered solution pH 7.4 (PBS, Gibco 10010023) to eliminate floating dead cells. Then, cells were trypsinized and fixated using 2% paraformaldehyde (Sigma Aldrich P6148) for 15 min at room temperature. Once fixed, cells were treated with the anti-human calreticulin antibody and PI (1 µg/mL) for 1 h at 37 °C in the dark. Later, the APC fluorescence was quantified in PI negative cells using a BD Accuri™ C6 flow cytometer (BD Biosciences), which was proportional to the amount of calreticulin exposure in the plasma membrane in live cells. Results were expressed as the fold increase in the signal compared to the untreated control.

### 4.9. ATP Measurement

At the end of the cytotoxicity assay, centrifuged supernatant was collected to measure ATP secretion. For this purpose, we used an ATP determination kit (Invitrogen™ A22066). Samples were analyzed in parallel to an ATP calibration curve (1, 5, 100, 250, 500 and 1000 µM). The tests were performed as recommended by the manufacturer. Results were expressed as the fold increase in the signal compared to the untreated control.

### 4.10. Perturbation-Based Mathematical Modeling

“A basic topology of the autophagic flux was designed in Cell Designer software (ver 4.4), which we later exported in SBML format to be used in Copasi (ver 4.24). The model consisted of a single compartment and 2 core species, including the phagophore at the cytoplasm (CYT), the autophagosomes (AP), the autophagolysosomes (AL) and the degraded ALs (sink). We imported the model in COPASI [68] and included species for each of the inhibitors fixing their activity as modifiers of the corresponding steps reported in the literature (see introduction) to reduce our degrees of freedom. We included 3 reactions: AP formation, conversion of APs to ALs and autophagolysosomal degradation (sink) including functions with mass-action kinetics including rate parameters and modification parameters for each perturbing agent. Then we proceeded to create a family of mathematical models for each cell line, in which each member of the family represented a possible combination (alternative hypothesis) of the possible modes of action of each chemotherapeutic drug on each of the reactions of the autophagic flux. Afterward, the results obtained in the autophagic flux kinetic tests were incorporated as experimental data and the species concentrations were coded as 1/0 to respectively indicate the presence/absence of a particular perturbation for each experimental condition. The parameter estimation was set as Time Course simulations with a Weight Method of Standard Deviation and the optimization method was set to Particle Swarm with an iteration limit of 2000, a swarm size of 50 and a standard deviation of 1 × 10^−6^. Finally, we proceeded to adjust all the generated mathematical models using that programmed Parameter Estimation task to obtain all the objective values for each proposed model. The hypothesis was ranked according to their degree of fitting to the experimental data to select the model with the highest likelihood to explain the specific mode of action of chemotherapy over the autophagic flux in each cell line”.

### 4.11. Statistical Analysis

The data represented in heatmaps corresponds to the median value of the performed replicates. For the cytotoxicity quantifications, results were expressed as the mean value ± standard deviation (SD). The differences between groups were analyzed by a one-way variance (ANOVA). Differences at *p* < 0.05 were considered statistically significant. Three independent replicates were conducted for each experiment.

## Figures and Tables

**Figure 1 metabolites-11-00637-f001:**
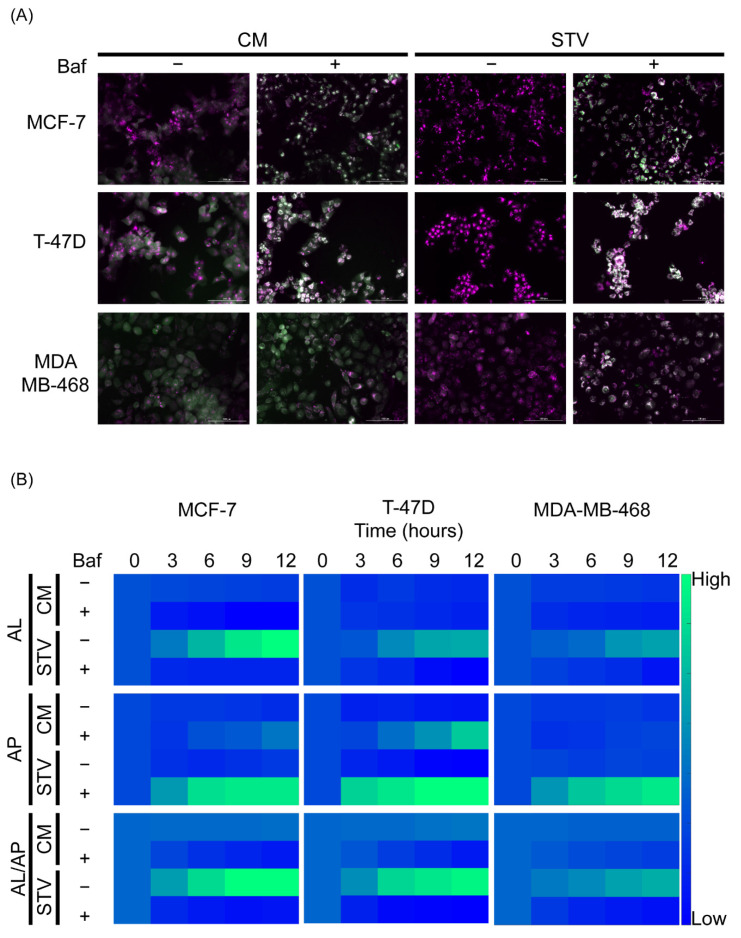
An mCHR-GFP tandem LC3 sensor reports dynamic changes in response to starvation-induced activation and bafilomycin-mediated degradation blockade. (**A**) Qualitative variations in the intensity and distribution of the mCherry (magenta) and GFP (green) fluorescence in three breast cancer cell lines stably expressing the autophagy sensor upon known autophagy perturbations like starvation and bafilomycin induced blockade. To emphasize the differences, only the images corresponding to 12 h post-treatment are shown. (**B**) The dynamic changes in the autophagy sensor were quantified using an image analysis pipeline designed with Cell Profiler software (Appendix A). mCherry+/GFP- puncta were measured as autophagolysosomes (AL), while mCherry+/GFP+ puncta were measured as autophagosomes (AP). Baf; bafilomycin, CM: complete medium, STV: starvation medium.

**Figure 2 metabolites-11-00637-f002:**
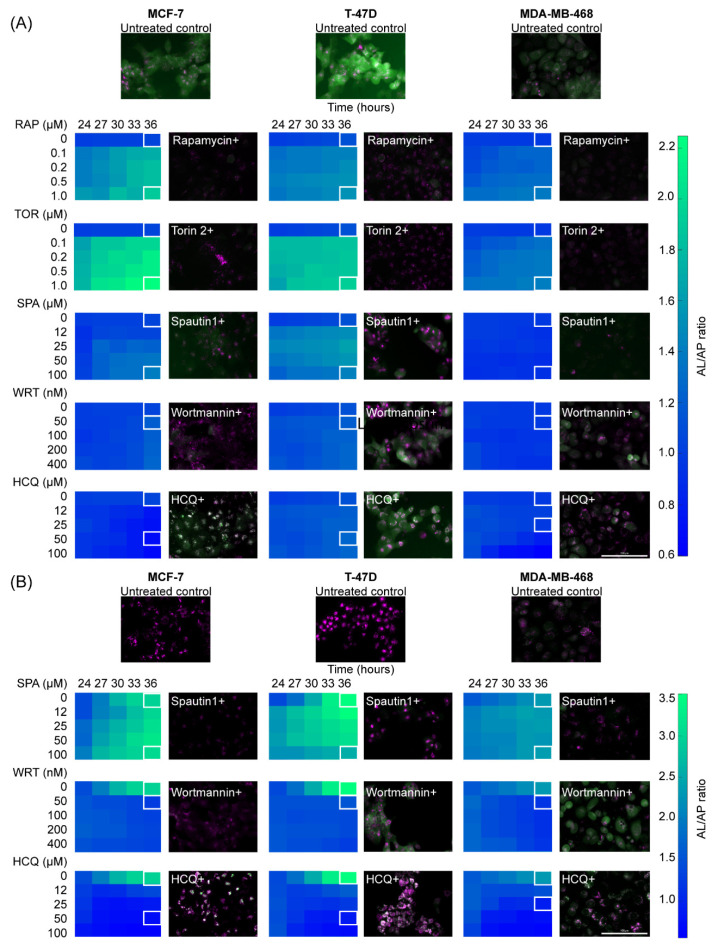
Breast cancer cell lines display a heterogeneous response to perturbations of the autophagic flux. (**A**) Variations in the autophagy sensor were monitored in three breast cancer cell lines upon different autophagy regulators treatment in the supplemented medium. RAP, TOR and SPA effect over the autophagy flux was measured after 24 h of treatment of the respective drug, then the medium was replaced with new supplemented culture medium containing the respective same drug concentration. WRT and HCQ activities over the autophagy flux are immediate, for that reason their effect over this pathway was quantified immediately upon treatment. (**B**) Variations in the autophagy sensor were monitored in three breast cancer cell lines upon different autophagy regulators treatment in starvation medium. SPA treatment began 24 h before the kinetic assay, then the supplemented medium containing this treatment was replaced with a starvation medium containing the same drug dose. No pretreatment was performed with WRT and HCQ. HCQ: hydroxychloroquine, RAP: rapamycin, SPA: Spautin-1, TOR: Torin-2, WRT: wortmannin.

**Figure 3 metabolites-11-00637-f003:**
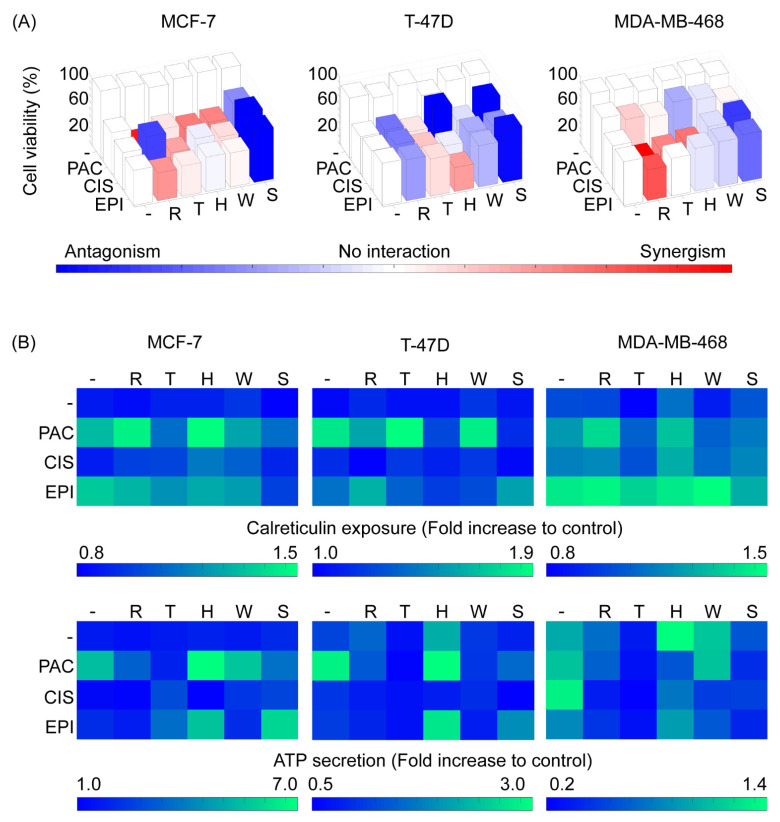
Cell line-specific profiles of cytotoxicity and cell death immunogenicity arise from the interactions between autophagy regulators and chemotherapeutic perturbations. (**A**) Cytotoxicity profiles for each chemotherapeutic drug and autophagy regulator combination for each cell line. Single perturbations are represented as white bars, while double perturbations are colored from blue (strong antagonism) to red (strong synergism) passing through white (no interaction). (**B**) Calreticulin exposure upon 24 h and ATP secretion upon 72 h of single and combined treatments. CIS: cisplatin, EPI: epirubicin, H or HCQ: hydroxychloroquine, PAC: paclitaxel R: rapamycin, S: Spautin-1, T: Torin-2, W: wortmannin.

**Figure 4 metabolites-11-00637-f004:**
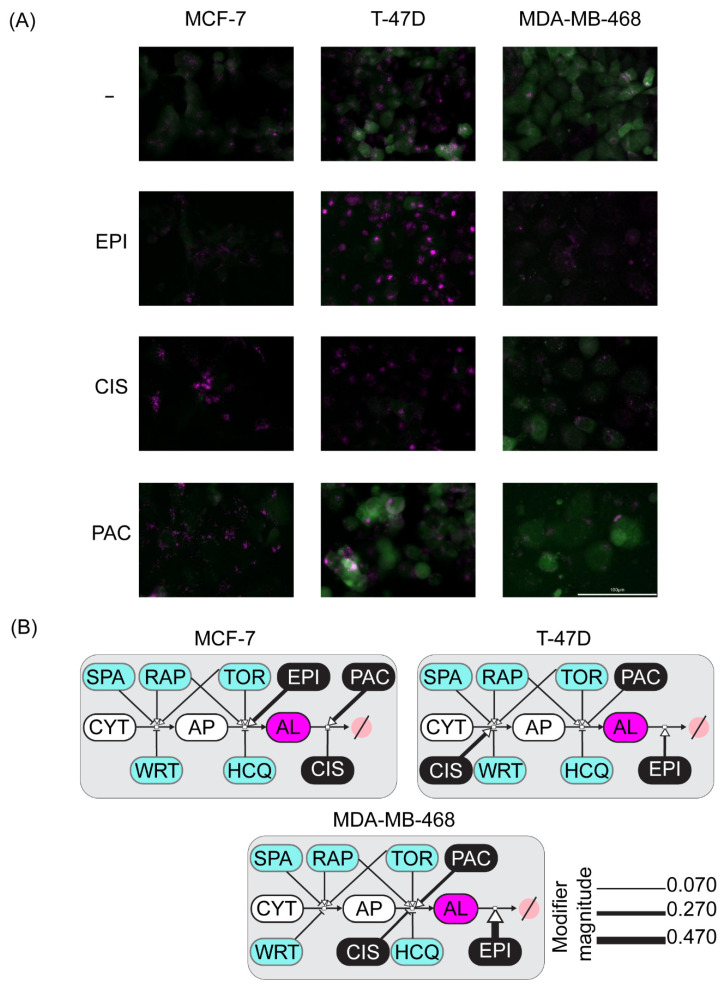
Perturbation-based mathematical modeling of the autophagic sensor dynamics indicates cell-line specific modes of action of chemotherapeutic agents on the autophagic flux. (**A**) Qualitative changes in the autophagy sensor induced by chemotherapeutic drugs treatment, to emphasize the differences, only the images corresponding to 48 h post-treatment are shown. (**B**) Cell-line specific modes of action of chemotherapeutic agents on the autophagic flux. A simplified topology of the autophagic flux was used. The magnitude of the modulating effect for each drug proposed activity is proportional to the line width. AL: autophagolysosomes, AP: autophagosomes, CIS: cisplatin, CYT: cytoplasm, EPI: epirubicin, HCQ: hydroxychloroquine, PAC: paclitaxel RAP: rapamycin, SPA: Spautin-1, TOR: Torin-2, WRT: wortmannin.

## Data Availability

The data presented in this study are available within the article and Appendix A.

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
