# Peer review of "Perturbation-Based Modeling Unveils the Autophagic Modulation of Chemosensitivity and Immunogenicity in Breast Cancer Cells"

_metabolites, 2021, doi:10.3390/metabo11090637_

Round 1
Reviewer 1 Report
Quirós-Fernández and collaborators aim to describe in this manuscript the effect of combining chemotherapeutic agents and autophagy modulators in the survival and cell death immunogenicity of different breast cancer cell lines. Moreover, in order to analyze all the resulting data at once, they develop and optimize a system biology approach by designing a perturbation-based mathematical model.
Overall, the goal of the study (which is well-written) is indeed necessary and interesting, as we need to know how autophagy regulators and cancer drugs interact if we want to target autophagy when treating this pathology. Furthermore, the authors clearly put a lot of effort into trying different conditions. Finally, I think the systems biology approach is very intriguing.
However, some major points should be addressed before it could be considered for publication:
1) One of my main concerns is the way results are presented.
On one hand, the quality of the images of immunofluorescence microscopy is not good enough, as puncta or even scale bars cannot be seen anywhere when you magnify the pictures on the screen. In fact, pictures look like plain black boxes some of the time. However, this may be due to the way pictures are compressed when preparing the file or similar. Better images should be shown. Also, some of the images/graphs are difficult to visualize. For example, images in Figures 2 and 4 are too small (besides their bad quality), and perhaps a higher magnification, showing fewer but bigger cells, should be used. Figure 4B is especially worrisome; the labels and rows are so thin and small that the analysis of the data is laborious and almost impossible, to the point that the figure is almost pointless. The authors should improve these figures so readers can easily follow the study (and enjoy it).
On the other hand, while the gradient colour scale is pleasant to see, I do not think using it to show AL/AP ratio is recommendable when analyzing autophagy. In fact, I believe including (at least as supplementary information) bars showing real numbers of AP and AL per cell would be much more informative in some cases, so the readers can see if an increase in the ratio is due to an increase in AL or a decrease in AP; or even to consider if the numbers are significant from a biological point of view. In fact, the authors talk throughout the paper about puncta quantification, but we never get to see the numbers, which are important to realize the scale of autophagy induction/repression in the system.
2) Use of autophagy modulators can be tricky, as the same compound can both repress or enhance the pathway depending on the extension of the treatment. For this reason, short periods of incubation are normally recommended. I understand that long treatments are used in this paper, but it may be important to discuss this issue in the paper, as it means that inducing/repressing effects of these compounds should not be taken for granted in advance.
3) A better, more detailed version of the method that the authors used to design the perturbation-based mathematical model should be included, as I find the current form to be quite vague. Also, in relation to point #2, and if I interpret the COPASI file correctly, the effect (or position) of autophagy modulators was set by default, not obtained from the data. Is that correct? Given the possibility that these modulators are not totally behaving as expected (because of the extended treatments), perhaps it would be interesting to also interrogate/test the model for the mode of action of these chemicals, and check where they would be placed in the model according to the experimental data obtained by the authors in their experiments.
4) I do not understand the "Statistical analysis" description in the "Materials and Methods". I may be wrong, but I do not see any graph with error bars or any statistical comparison with p-values. What is this section referring to?
Minor points:
a) I think there is a typo in line 79 ("process").
b) Is there any specific reason why these three breast cancer cell lines were used? MCF-7 and T-47D are classified as "luminal", while MDA-MB-468 is considered "basal", having different immune and molecular characteristics that can for sure determine their response to the treatments. I think it would be interesting to include some discussion about this in the manuscript.
c) In line 196, the authors talk about "yellow fluorescent puncta". However, in the version of the pictures that I have, I see a white signal instead of a yellow one. This should be clarified to avoid confusion (either changing the text or adjusting the pictures).
d) In lines 352-353, the authors talk about changes in the size of autophagic vacuoles and the formation of clusters. How was this measured/analyzed? I think it would be interesting to include these data/images, with their corresponding statistical analysis.
d) In line 439 there is an extra space in "ATG7".
e) In the same paragraph (lines 438-447), the authors explain the action of the two ubiquitin-like conjugation systems (ATG12-ATG5 and ATG8). However, given that the article focuses on human autophagy, I suggest they clarify that LC3 is only one of the main subfamilies of ATG8-like proteins in humans (LC3 and GABARAP) and that there are four different ATG4 proteases (ATG4A-D).
f) There is a typo in line 581 ("DMSO")
g) Line 627: A separation from the previous section is missing.
h) This may be a problem on my side, but while I am able to open the COPASI file, I get an error when I try to open the CellProfiler pipeline file. I recommend double-checking every file in case their publication is required by the journal.
i) Is there any way the model could be a posteriori validated by the authors? Even though it is the result of analyzing a large amount of data, an additional validation of the model with a new experiment would greatly improve the work (and its validity). Could the authors at least suggest how the model could be validated?
j) I also encourage the authors to discuss any idea on how the same chemotherapy agent can have different effects on the same pathway (autophagy) in different cell lines.
Reviewer 2 Report
Autophagy is induced by almost all conventional treatments of breast cancer and is considered a target for pharmacologic blockade in the clinic. Autophagy is largely accepted as a pro-survival mechanism in tumor cells and has generated significant interest in cancer research and treatment strategies. Hoeever, it plays multiple and often disparate roles during different stages of tumorigenesis and in response to anti-tumor treatments. Consequently, it is important to further our understanding of authopgahic processes and their role in breast cancer. In the manuscript “Perturbation-based modeling unveils the autophagic modulation of chemosensitivity and immunogenicity in breast cancer cells” Quiros-Fernandez and coauthors investigated the association between autophagy regulation and the exposure of DAMPs to immunogenic cell death. The current study is technically convincing and the overall claim is supported by the results. The authors used three different breast cancer cell lines to explore the interaction between autophagy modulators and chemotherapeutic agents in regards of cytotoxicity and ICD. Furthermore, they designed a perturbation-based model in which they propose specific modes of action of chemotherapeutic agents on the autophagic flux. After careful evaluation, I have some suggestions for the current version of the manuscript, which may require the authors’ further consideration:
Major:
- As already mentioned, autophagy plays a paradoxical role in tumorigenesis, depending on the stage of tumor development. Early in tumorigenesis, autophagy is a tumor suppressor via degradation of potentially oncogenic molecules. In advanced stages, however, autophagy promotes the survival of tumor cells by ameliorating stress in the tumor microenvironment. These controversial roles of autophagy are due to its involvement in diverse cellular pathways. Recent evidence suggests that autophagy is activated in different stages of the metastasis process. Maybe autophagy may be the mechanism responsible for metastatic cancer cells undergoing adaptation to escape all adverse environments. Thus, I am wondering why the authors only investigated dormant or proliferative cell lines and not included a highly metastatic one such as MDA-MB-231, MDA-MB-435, SUM149 or BT-549. Authors should at least discuss their selection of the cell lines used in the study.
- Although measuring autophagy flux with specific proteins undergoing autophagic degradation of LC3 could provide a precise evaluation of the autophagic activity, it should be mentioned that some residual autophagy is independent of LC3 (doi:10.1016/j.ymeth.2014.11.021, doi:10.1016/j.cell.2010.01.028).
Minor:
- 3. L38: The reference for breast cancer incidence is out of date. Please refer to a more recent article such as doi:10.3322/caac.21660 or doi:10.1002/ijc.33588.
- 4. Figures 1,3,4: Please use the complete description of the cell type “MDA-MB-468” for labelling.
- 5. Methods/Statistics: Please think about using SEM. The standard error of the mean only indicates how well you determined the quantity sought, but only standard deviation will show the real variation in your measurements.
Reviewer 3 Report
The paper entitled “Perturbation-based modeling unveils the autophagic modulation of chemosensitivity and immunogenicity in breast cancer cells” that you kindly submitted for publication in the “Metabolites” now been considered.
In the manuscript, the authors have developed new model to explore and analyze the cellular dynamics of autophagy by live imaging two different fluorescence proteins in cells. A variety of stimuli was applied to induce autophagic process and relative ratio between AP and AL by measuring and analyzing fluorescence, GFP and mCherry in the presence of absence of bafilomycin. Furthermore, they further applied their system on cells treated with several cytotoxic drugs such as cisplatin, epirubicin, paclitaxel with combination with autophasic stimuli. This is interesting and fine to sense the autophagic flux in cancer cells in response to chemotherapeutic agents. However, there are a few things to be done before publication.
Specific comments:
- In Figure 1 and 2, three cell lines went through starvation in the presence or absence of bafilomycin timely and analyzed for AP, AL and their ratio. It is good to start to check the stability of fluorescent proteins in different situations like pH during autophagic process. One thing is that this study is trying to suggest another way to analyze and get information for autophagic flux and autophagy in this case should be validated precisely. For this, it should be strengthened to show autophagic proteins in each step parallelly at protein levels as well as autophagy in cells.
- In Figure 3, the correlation between cytotoxicity and autophagic flux might be shown in table with significance please
- It will be helpful to show simplified working model for better understanding.
Round 2
Reviewer 3 Report
now the revised manuscript seems to be much improved for the publication